# Dispositional Awe Positively Predicts Prosocial Tendencies: The Multiple Mediation Effects of Connectedness and Empathy

**DOI:** 10.3390/ijerph192416605

**Published:** 2022-12-10

**Authors:** Liming Jiao, Li Luo

**Affiliations:** 1The Department of Economics and Management, Neijiang Normal University, Neijiang 641112, China; 2The Department of Education Science, Neijiang Normal University, Neijiang 641112, China; 3Faculty of Psychology, Southwest University, Chongqing 400715, China

**Keywords:** dispositional awe, connectedness, empathy, prosocial tendencies, multiple mediation

## Abstract

Awe is an emotion frequently experienced by individuals in different cultures. When individuals experience awe, they would feel a sense of connectedness to other people or nature arises, and shift their attention to the outside world, which would increase empathy for others in need and, in turn, improve their prosocial tendencies. To test this proposal, we applied a cross-sectional study using a questionnaire survey to collect a sample of 1545 (*N*_female_ = 988) in Asia, aged between 16 and 71 years old (*M* = 22.81, *SD* = 7.80). The Structural Equation Model and bootstrapping method were used to test the mediation effects of connectedness and empathy between awe and prosocial tendency. Results showed that dispositional awe positively predicted a prosocial tendency, which could be partially explained by the multiple mediation effects of connectedness and empathy, after controlling for the effect of the small self. The findings deepen researchers’ understanding of the underlying mechanisms of the relationship between awe and prosociality and have practical implications for cultivating awe and prosocial behavior.

## 1. Introduction

Schneider [1] believes that awe plays an important role in the healthy development of the individual and society. Awe is an important research topic in positive psychology, which refers to an emotion when individuals encounter vast and powerful stimuli that are beyond their own understanding, and they should accommodate to the current context [2]. There are many kinds of elicitors that induce awe, such as social elicitors (e.g., powerful leader, religion), physical elicitors (e.g., tornado, grand vista), and cognitive elicitors (e.g., grand theory) [2]. As a collective and self-transcendent emotion, awe encourages individuals to go beyond their own momentary desires and enhance the welfare of others [3,4,5], which enables them to fold into collaborative social groups and engage in collective action [6].

A body of related studies demonstrates that awe is a key promoter to prosocial actions [4,7,8], such as self-sacrifice, donation, cooperation, and sharing [9]. On the trait level, individuals with a higher awe tendency are more prosocial [10,11,12]. Participants scoring higher on the dispositional awe scale were allocated more tickets to the partner in the dictator game [4] and were also more willing to provide support, guidance, and sharing to help others online [13]. They are more prosocial [14]. In addition, laboratory studies also demonstrated that state awe played an important role in promoting prosocial behavior. Compared with participants in neutral or other positive emotions groups, participants in the awe group were more likely to help others [4,7,8,15]. Piff, Dietze [4] indicated that participants who spend one minute looking up at the tall trees would trigger the experience of awe, which in turn promoted them to help the experimenter to pick up more pens dropped on the ground. Yang, Yang [15] found that compared with the experience of happiness and neutrality, awe could promote participants to choose more simple tasks for their partners to help them win. In all, awe serves a crucial social function to perform altruistically.

How does awe enhance prosociality? Although precious studies found that awe could predict prosocial behavior via the sense of the small self, a relatively diminished sense of self [4,16]. This account has not received consistent supporting evidence [15], which suggests that the small self may not be a reliable mediator. That is, more reasonable and reliable pathways underlying the association are waiting to be investigated. In the present study, we proposed that connectedness and empathy could be the mediators of the association between awe and prosocial tendency. Prior to presenting our empirical evidence, we will elaborate on our rationale.

The feelings as information theory proposes that individuals’ feelings can be the source of information which would affect subsequent judgments and decisions, and different emotions produce different feelings [17]. The sense of connectedness is the main feeling produced by awe [3], which is defined as the feeling of unity with society, nature, humanity, and even with the universe beyond one’s self feature [18,19]. A growing body of evidence shows that individuals could experience a deep sense of connectedness when they are feeling awe [3,18,19,20,21]. Findings from the study of phenomenology have shown that people always feel connected with the universe, the divine, or something nonspecific when experiencing awe [3]. Experimental research also indicates that awe improves people’s integration with the world and others [19]. How individuals perceive the connectedness with other people, the world, and nature may affect their behaviors. There is converging evidence that connectedness could promote prosociality. Vingerhoets, van de Ven [22] found that people with a higher sense of social connectedness were more likely to help others. Even for the infants, connectedness with the experimenters manipulated by interpersonal motor synchrony would predict the subsequent helping behavior [23]. Additionally, those who perceived a stronger connectedness with nature were more likely to engage in pro-environment behaviors [24,25,26]. All of these findings imply that connectedness can promote individuals’ prosocial behavior and it may mediate the link between awe and prosocial behavior.

Moreover, as a positive emotion, awe may also predict prosocial behavior through empathy. Empathy, a multidimensional concept that includes experience sharing, mentalizing, and empathy concern [27], plays an important role in motivating many forms of prosociality. The empathy–altruism hypothesis proposes that individuals are likely to help others if they have high empathy levels [28]. There are many empirical studies that have shown that individuals with higher traits of empathy are more prosocial [29,30,31]. It was found that when watching strangers being rejected in a social exclusion game, participants with higher levels of empathy had higher activation of brain areas associated with empathy (e.g., insula and medial prefrontal cortex), which predicted subsequent prosocial behavior toward the victims [30]. Meta-analytic evidence also suggested the promoting effect of empathy on prosocial behavior [32].

The broaden-and-build theory of positive emotions states that positive emotions could share the ability to broaden individuals’ momentary thought–action repertoires and build their enduring personal resources, ranging from physical and intellectual resources to social and psychological resources, which provides possibilities to support others [33]. A study showed that awe induced by the classical serotonergic psychedelic increased feelings of empathetic drive which mediated the relationship between awe and exploitative-entitled narcissism [34], which suggested that experiencing awe improved empathy. Moreover, evidence from functional magnetic resonance imaging studies also discovered the similarity of awe and empathy in terms of neuromechanisms that both of them reduced the activation of the default mode network and enhanced the activation of the externally directed frontoparietal control network [35,36]. Therefore, empathy may subserve the prediction from awe to prosociality.

Studies also confirmed that connectedness positively predicted empathy [37]. When participants perceived higher levels of connectedness or self-integration with others, they were more empathic toward others’ sufferings [30,38,39]. On the contrary, stressing the distinction between self and others may decrease empathy [11]. Fido and Richardson [37] found that connectedness was positively associated with empathy and inversely associated with callous and uncaring, and empathy mediated the relationship between connectedness and both callous and uncaring traits. In addition, evidence from clinical practice research discovered that individuals increased their empathy after receiving human–animal bond training [40].

Although previous studies have explored the relationship among dispositional awe, connectedness, and prosocial behavior. No study has examined the serial mediation effect of connectedness and empathy. The exploration of this question could deepen our understanding of the relationship between awe and prosocial tendencies. Therefore, the purpose of this study is to investigate the mechanisms by which dispositional awe predicts prosocial tendencies from the perspective of connectedness and empathy. We hypothesized that connectedness could predict empathy, and the two variables may serially mediate the association between awe and prosocial tendencies. The proposed media model was shown in Figure 1. Using a cross-sectional design, we tested our proposal.

## 2. Materials and Methods

### 2.1. Study Sample and Data Collection

The study was conducted in accordance with the Declaration of Helsinki, and approved by the internal ethics committee of the Faculty of Psychology in Southwest University (Protocol code H20081). Informed consent was obtained from all subjects involved in the study verbally. We used random sampling to recruit students from Neijiang Normal University in China, and asked them to invite their classmates or families to participate through the snowball sampling method. The subjects could receive credits after completing the survey or inviting others to fill in the questionnaires. To ensure the validity of the data, we informed them in advance that they could obtain credits after the validity of the data was approved by the researchers. There were 1717 participants who completed the online survey. The data of 170 participants were removed from further analysis due to (1) data being repeatedly submitted, and (2) taking quite a short time to finish the whole survey (<100 s). Therefore, the available data was 1545 (*N*_female_ = 988). The age ranged from 16 to 78, with an average age of 22.81 ± 7.80 years.

### 2.2. Materials

#### 2.2.1. Dispositional Awe

The dispositional awe was assessed by the awe sub-scale in Dispositional Positive Emotion Scale (DPES-awe) [41]. DPES-awe is a unidimensional scale, consisting of 6 items, such as “I often feel awe”. Participants were asked to respond to each item on a 7-point scale (1 = strongly disagree to 7 = strongly agree). A higher score indicates individuals experience the emotion of awe more often. Cronbach’s alpha of the scale was 0.90.

#### 2.2.2. Prosocial Tendencies

The Prosocial Tendency Measure (PTM) was used to assess prosocial tendencies [42]. The 26-item scale has six sub-scales: public, anonymous, dire, emotional, compliant, and altruism. Each sub-scale is an assessment of prosocial tendencies in the specific context. Participants responded to the items on a 5-point scale (1 = does not describe me at all, 5 = describes me greatly). The higher the score, the more likely the participants is to perform prosocial behavior in this context. Cronbach’s alpha of each sub-scale ranged from 0.67 to 0.82.

#### 2.2.3. Connectedness

Connectedness was assessed by 5 items which were developed for this study (“I think I belong to a larger entity”, “Human beings is a whole”, “Human beings should not fight against each other”, “Human beings is a community with a shared future”, “Human beings is intimately connected to all living things”). Participants reported their experience on a 7-point scale (1-not at all, 7-very much). A higher score indicates higher experience on connectedness. To ensure the reliability and validity of this study, half of the data was used as exploratory factor analysis (EFA) and the other half as confirmatory factor analysis (CFA). The results of EFA showed that this scale could extract one factor with a characteristic root greater than 1 and the explanatory rate was 64.64%. CFA found that the factor loads of the five items were all significant, ranging from 0.60 to 0.89, and the fitting index was also acceptable, χ^2^ (3) = 11.94, GFI = 0.99, AGFI = 0.97, NFI = 0.99, CFI = 0.99, RMSEA = 0.06, SRMR = 0.02. The Cronbach α coefficients of the two parts were 0.83 and 0.85, respectively.

#### 2.2.4. Empathy

The Interpersonal Reactivity Index (IRI) was employed to assess participants’ empathy [43]. This scale consists of 22 items and has 4 sub-scales: personal distress, perspective taking, fantasy and empathic concern. Participants should respond to each item on a 5-point scale (0 = does not describe me at all, 4 = describes me greatly). The higher the score, the more empathic to others. Cronbach’s alpha of each sub-scale ranged from 0.67 to 0.82.

#### 2.2.5. Small Self

Because there was some evidence that showed the sense of the small self played a mediation role between awe and prosociality [4], we measured it to control its effect. It was assessed by two items, “In general, I feel relatively small” and “In general, I feel insignificant” [4], using a 7-point Likert scale, ranging from 1 (not at all) to 7 (very much).

### 2.3. Data Analysis

Descriptive statistics and Pearson correlation coefficients were conducted in SPSS 23.0, and path analysis was estimated in AMOS 23.0 (SPSS Inc, Chicago, IL, USA). All data have been made publicly available via The Open Science Framework repository, named data for dispositional awe and prosocial tendencies, and can be accessed at https://osf.io/rfyt5/ (accessed on 3 October 2022).

## 3. Results

### 3.1. Common Method Variance

Because the present study is a cross-sectional research design, that is, all of the data were self-reported and collected during the same period of time, a common method bias (CMB) may exist [44]. We examined CMB using Harman’s one-factor method. The results of the principal component analysis showed that all of the variables produced seven distinct factors with characteristic roots greater than one, which accounted for 58.98% of the total variance. The first factor accounted for 30.47% of the variance, lower than 40%. Therefore, CMB is not serious in the present study.

### 3.2. Descriptive and Correlated Analysis

The descriptive statistics of the variables and Pearson’s correlation coefficient to define the relationships among the variables are presented in Table 1.

The results showed a significant pairwise correlation among variables. That is, dispositional awe was positively correlated with a sense of connectedness, empathy, and prosocial tendencies. Connectedness and empathy were positively correlated with prosocial tendencies.

### 3.3. Mediate Analysis

The Structural Equation Model (SEM) and a parametric bootstrap procedure with 5000 replications were used to calculate 95% bias-corrected CIs for the indirect effects of the parameters and standard errors. If the confidence interval contains 0, it indicates that the mediation effect is insignificant, otherwise, the mediation effect is significant. The model was evaluated using the χ^2^ statistic and approximate fit indices. The model included dispositional awe as the input variable, connectedness and empathy as the mediators, each dimension of prosocial tendencies as the outcome variables, and the sense of the small self as the control variable. The path coefficients and effect decomposition of the models are shown in Table 2 and Table 3, respectively.

The results of the path analysis showed that the total and direct effects from dispositional awe to prosocial tendencies in each specific context were significant. The interval estimates of the mediation effects of connectedness and empathy alone did not contain 0, except that connectedness alone could not significantly mediate the link between dispositional awe and public prosocial tendencies. We found the significant serial mediation effect of connectedness and empathy, which illustrated that connectedness and empathy serially mediated awe’s positive predictive effect on prosocial tendencies. Using a pairwise comparison of the strength of the three mediation effects, we found that the mediation effect of empathy was stronger than the serial mediation effect between awe and each type of prosocial tendencies, *p* < 0.001, and the mediation effect of connectedness was stronger than the serial mediation effect between awe and prosocial tendencies under the contexts of altruism, anonymity, and emergency, *p* < 0.018. In addition, the effect of empathy was stronger than that of connectedness in the public contexts *p* = 0.001. other pairwise comparisons were insignificant, *p* > 0.058. In addition, the mediation effect of the small self was not significant. All of the mediation effect analysis suggested that the two mediators, connectedness, and empathy, could account for the relationship between dispositional awe and prosocial tendencies, even controlling for the effect of the small self.

## 4. Discussion

There has been some research exploring the effects of awe on prosocial behavior and the mediation role of the small self [8,20]. In the present study, we investigate the relationship between awe and prosocial tendencies and the mechanisms from the perspective of connectedness and empathy, controlling for the effect of the small self. We discovered that awe could positively predict prosocial tendencies, and this result was partially mediated by connectedness and empathy, but not by the sense of the small self.

Previous studies found that awe could increase prosocial behavior, such as ethical decision-making, generosity, prosocial value, and helping behaviors [8,20]. In line with these studies, the present work also confirms that the experience of awe positively predicts prosocial tendencies in different contexts. The hyper-altruistic theory proposes that people would pay attention to the interests of others when they pursue their own interests. If there is a conflict between self-interest and others’, hyper altruism can form a cognitive model of “others’ interests > self-interest”, thus influencing the later behavioral decisions [45,46]. Awe is a kind of self-transcendent emotions, which encourages individuals to transcend their own momentary needs and desires and to focus on those of another. Therefore, awe can form this cognitive pattern to make individuals pay more attention to others’ interests and increase prosocial behavior [47].

We observed the mediation effects of connectedness and empathy on the relationship between awe and prosocial tendencies. This effect was still robust after isolating the potential effect of the small self. The feelings in the information theory proposes that feelings can be the source of information to affect individuals’ judgments and decisions [17]. The sense of connectedness is the main feeling brought by awe [3], which could act as the information to influence subsequent behaviors. Previous studies showed that connectedness could promote one’s prosociality in many forms [26,39,40,48,49]. A recent study found that connectedness with nature played a mediation role between awe and ecological behavior [15]. Another study also has shown that connectedness could interpret the negative predictive effect of awe on maladaptive narcissism, a tendency of grandiose fantasies and the need for admiration which may reduce concern for others’ welfare [50]. All of these lend support to the current findings. In addition, studies revealed that positive emotions may inspire empathy, which in turn enhanced prosocial behavior [30,31,39,51]. We found that the mediation effect of empathy was significant in the present study, which confirmed the broaden-and-build theory of positive emotions.

This study also discovered the significant serial mediation of connectedness and empathy. The findings of previous studies suggested that connectedness could positively predict empathy, which jointly predicted prosocial behavior [37]. The evidence from clinical practice also found that human–animal bond training can increase individuals’ empathy [40]. Connectedness reflects the sense of belonging and intimacy with the outside world. Perceiving connectedness with others helps the increasing empathy for others [11,38,39]; in contrast, emphasizing the distinction between self and others can reduce empathy [11]. Twenge, Abebe [52] believed that prosocial behavior was based on the sense of belonging. Therefore, the significant serial mediation is reasonable.

We did not observe the mediation effect of the small self, which is contrary to the results of existing studies. It is found that the sense of the small self could explain the effects of awe induced by a natural scene on prosocial behavior [4,15]. Although many studies suggested that the small self is an important feeling when someone experiences awe [4,53], it may be not necessary to trigger prosocial behavior. In the present study, the small self could not predict prosocial tendencies in any context. Le, Saltsman [54] discovered that awe could not reduce stress because awe may generate different types of the small self, which affected the subsequent performance diversely. Awe led participants who take on a self-distanced perspective to assess stressors more manageable, but led those who take on a self-immersed perspective to assess stressors that are less manageable. This suggests that we need to consider the effects of different types of the small self on prosocial behavior or other response variables in the future.

The present work contributes to our understanding of the relationship between awe and prosocial tendencies, and also provides practical implications concerning the cultivation of prosocial behavior from the perspective of awe. However, we should be cautious to infer the causality, because this is a cross-sectional study using the questionnaire survey. Further research is needed to replicate these findings with longitudinal and/or experimental approach. In addition, empathy is a multi-dimensional concept, including emotional empathy and cognitive empathy [27]. Previous studies have found that different components of empathy have different effects on prosocial behavior [31]. Therefore, whether cognitive and emotional empathy play different roles in the relationship between awe and prosocial behavior also needs further investigation.

## 5. Conclusions

Previous studies have examined the mechanism of awe on prosocial behavior from the perspective of the small self, but have not found consistent results. Based on the feelings as information theory and the broaden-and-build theory of positive emotions, this work examined the mediation role of connectedness and empathy in awe on prosocial behavior. It is found that awe positively predicts a sense of connectedness with society and the world, which in turn links with empathic concern for other people and prosocial tendencies. These results exist independently even after controlling for the effect of the small self. The present investigation reveals the impact of awe on prosocial tendencies and the mechanisms, which extends our understanding of the mechanisms underlying the relationship between awe and prosociality.

## Figures and Tables

**Figure 1 ijerph-19-16605-f001:**
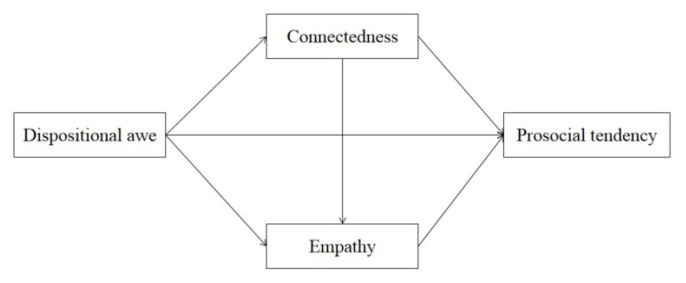
The proposed media model.

**Table 1 ijerph-19-16605-t001:** Descriptive statistics and correlations among the variables.

Variables	*M* ± *SD*	1	2	3	4	5	6	7	8	9
1. Awe	5.34 ± 1.16	1								
2. Connectedness	5.84 ± 1.02	0.42 ***	1							
3. Empathy	2.46 ± 0.47	0.37 ***	0.38 ***	1						
4. Emotional	3.84 ± 0.68	0.44 ***	0.39 ***	0.53 ***	1					
5. Compliant	3.79 ± 0.67	0.38 ***	0.36 ***	0.49 ***	0.83 ***	1				
6. Altruism	3.98 ± 0.71	0.43 ***	0.45 ***	0.50 ***	0.78 ***	0.76 ***	1			
7. Anonymous	3.80 ± 0.71	0.40 ***	0.35 ***	0.40 ***	0.73 ***	0.72 ***	0.75 ***	1		
8. Public	3.54 ± 0.79	0.34 ***	0.23 ***	0.39 ***	0.68 ***	0.67 ***	0.51 ***	0.55 ***	1	
9. Dire	3.98 ± 0.68	0.46 ***	0.44 ***	0.50 ***	0.77 ***	0.74 ***	0.75 ***	0.69 ***	0.60 ***	1

Note: *** *p* < 0.001.

**Table 2 ijerph-19-16605-t002:** The path analysis among the variables.

Independent Variables	Small Self	Connectedness	Empathy	Anonymous	Public	Compliant	Altruism	Dire	Emotional
Small self				0.01	0.02	0.002	−0.04	0.01	−0.02
Dispositional awe	0.23 ***	0.42 ***	0.26 ***	0.24 ***	0.22 ***	0.18 ***	0.20 ***	0.24 ***	0.24 ***
Connectedness			0.27 ***	0.15 ***	0.02	0.14 ***	0.25 ***	0.21 ***	0.16 ***
Empathy				0.25 ***	0.29 ***	0.37 ***	0.34 ***	0.33 ***	0.39 ***

Note: All path coefficients were standardized. *** *p* < 0.001.

**Table 3 ijerph-19-16605-t003:** Effect decomposition and interval estimation of each dimension of prosocial tendencies predicted by dispositional awe.

Effects	Anonymous	Public	Compliant	Altruism	Dire	Emotional
Total effect	0.42 *** [0.37, 0.46]	0.39 *** [0.33, 0.48]	0.38 *** [0.34, 0.43]	0.42 *** [0.38, 0.47]	0.46 *** [0.41, 0.50]	0.44 *** [0.40, 0.49]
Direct effect	0.24 *** [0.19, 0.29]	0.01 *** [−0.004, 0.02]	0.18 *** [0.14, 0.23]	0.20 *** [0.16, 0.25]	0.24 *** [0.20, 0.29]	0.24 *** [0.19, 0.29]
Indirect effect of small self	0.002 [−0.01, 0.01]	0.05 [−0.004, 0.01]	0 [−0.01, 0.01]	−0.01 [−0.02, 0]	0.001 [−0.01, 0.01]	−0.01 [−0.01, 0.003]
Indirect effect of connectedness	0.06 *** [0.04, 0.09]	0.01 [−0.01, 0.03]	0.06 *** [0.04, 0.08]	0.10 *** [0.08, 0.13]	0.09 *** [0.07, 0.11]	0.06 *** [0.04, 0.09]
Indirect effect of empathy	0.07 *** [0.05, 0.09]	0.08 *** [0.06, 0.10]	0.10 *** [0.08, 0.12]	0.09 *** [0.07, 0.11]	0.09 *** [0.07, 0.11]	0.10 *** [0.08, 0.12]
Serial mediation effect	0.03 *** [0.02, 0.04]	0.03 *** [0.03, 0.04]	0.04 *** [0.03, 0.05]	0.04 *** [0.03, 0.05]	0.04 *** [0.03, 0.05]	0.05 *** [0.04, 0.06]

Note: The data in [] is the 95% interval estimation. *** *p* < 0.001.

## Data Availability

All data and analysis code have been made publicly available via The Open Science Framework repository, named data for awe and aids, and can be accessed at https://osf.io/rfyt5/ (accessed on 3 October 2022).

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
