# Peer review of "Dispositional Awe Positively Predicts Prosocial Tendencies: The Multiple Mediation Effects of Connectedness and Empathy"

_ijerph, 2022, doi:10.3390/ijerph192416605_

Round 1
Reviewer 1 Report
Please describe the research methods or main data processing methods in the abstract
Author Response
Reviewer1
1. Please describe the research methods or main data processing methods in the abstract.
Response:
Thank you for pointing this out. According to your suggestion, we have described the research methods and main data processing methods in the abstract.
Text change:
Line 10-17
Awe is an emotion frequently experienced by individuals in different cultures. When individuals experience awe, they would feel a sense of connectedness to other people or nature arises, and shift their attention to the outside world, which would increase empathy for others in need and in turn improving the prosocial tendencies. To test this proposal, we applied a cross-sectional study using questionnaire survey to collect a sample of 1545 (Nfemale = 988) in Asia, aged between 16 and 71 years old (M = 22.81, SD = 7.80). The Structural Equation Model and bootstrapping method were used to test the mediation effects of connectedness and empathy between awe and prosocial tendency.

Reviewer 2 Report
Abstract should be a bit more general, to place the article into some research discipline. When reading just the abstract I did not know, if the article is about educational studies or psychology. It was unclear if connectedness and empathy refers on people’s relations to other people/environment/nature/. Also, the abstract could contextualize this research into some cultural context: is the research made in USA, Asia, Europe – for example. As a reader, I assume, that awe is culture-specific phenomena. – Anyhow the abstact should give better context for the study. On the other hand, the abstract does not need to tell about the method and materials so detailed as persented now. At the very end of the article, of good general description is made: "a cross-sectional study using questionnaire survey."
The article would benefit of a very brief introduction that would also place the study into some academic discipline and state the importance of the research topic as well as the research aim and research question. The concept of awe could be introduced better. For example, I was familiar with concepts of connectedness and empathy, but not awe, and reading your article makes me consider, if it is related to some religions orworld view. Is awe part of patriarchal and/or authoritarian society? Is it oppositional or in line with the direction that fosters criticality, emancipation, empowerment and agency of members of society and communities in society. – Currently you present some aspects of awe in the discussion part, for example line 228. The connectedness is explained even later, in the line 253.
The chapter “1. Introduction” is well written and gives the state of art to the research topic. References to previous literature are clear. Anyhow, I can recommend some recents articles and books about empathy and connecteddness.
Materials and methods – I am not expert of evaluating this type of methodology and its presentation.
In the discussion the sentence: “Consistent with our findings, a recent study found that people with greater dispositional awe-proneness were more willing to self-sacrifice for their group.” This makes me again wonder, in what kind of society this is a desired pattern of behaviour. In army? Coud this refer to suicide bombe?
In the paragraph starting from the line 234 you tell of positive impact of awe. But in my opinion, from line 242 you jump to impact that is not desired. Maladaptive narcissism is not hoped part of personality. This jump from good to bad should be somehow separated in the text.
Conclusion could be a bit long and expalain again why this research was made and how.
Possible reading
Raatikainen, K.J.; Juhola, K; Huhmarniemi, M. & Peña-Lagos, H. (2020) “Face the cow”: reconnecting to nature and increasing capacities for pro-environmental agency, Ecosystems and People, 16:1, 273-289, DOI: 10.1080/26395916.2020.1817151
Kravtsov, T.; Huhmarniemi, M. & Kugapi, O. (2022). Enhancing empathy in creative tourism. In M. Sarantou & S. Miettinen (Eds.), Empathy and business transformations (pp. 31–43 ). Routledge.
M. Sarantou & S. Miettinen (Eds.) (2022). Empathy and business transformations https://doi.org/10.4324/9781003227557
Author Response
Reviewer2
- Abstract should be a bit more general, to place the article into some research discipline. When reading just the abstract I did not know, if the article is about educational studies or psychology. It was unclear if connectedness and empathy refers on people’s relations to other people/environment/nature/. Also, the abstract could contextualize this research into some cultural context: is the research made in USA, Asia, Europe – for example. As a reader, I assume, that awe is culture-specific phenomena. – Anyhow the abstact should give better context for the study. On the other hand, the abstract does not need to tell about the method and materials so detailed as persented now. At the very end of the article, of good general description is made: "a cross-sectional study using questionnaire survey."
Response:
Thanks very much for these valuable suggestions. According to these suggestions, we have revised abstract in our manuscript with blue font.
Text change:
Line 10-17
Awe is an emotion frequently experienced by individuals in different cultures. When individuals experience awe, they would feel a sense of connectedness to other people or nature arises, and shift their attention to the outside world, which may be good for increasing empathy for others in need and in turn improving the prosocial tendencies. To test this hypothesis, we applied cross-sectional study using questionnaire survey to collect a sample of 1545 (Nfemale = 988) in Asia, aged between 16 and 71 years old (M = 22.81, SD = 7.80). The Structural Equation Model (SEM) and bootstrapping method were used to test the mediation effects of connectedness and empathy between awe and prosocial tendency.
- The article would benefit of a very brief introduction that would also place the study into some academic discipline and state the importance of the research topic as well as the research aim and research question. The concept of awe could be introduced better. For example, I was familiar with concepts of connectedness and empathy, but not awe, and reading your article makes me consider, if it is related to some religionsor world view. Is awe part of patriarchal and/or authoritarian society? Is it oppositional or in line with the direction that fosters criticality, emancipation, empowerment and agency of members of society and communities in society. – Currently you present some aspects of awe in the discussion part, for example line 228. The connectedness is explained even later, in the line 253.
Response:
We sincerely appreciate reviewer’s suggestion. According to this suggestion, on the one hand, we stated the importance of the research topic as well as the research aim and research question in the instruction. On the other hand, we have introduced the concept of awe and its elicitors in the section of introduction. In the field of psychology, awe is considered as an emotion, but not world view. Meanwhile, there are many kinds of elicitors to induce the emotion of awe, such as social elicitors (e.g., powerful leader, religion), physical elicitors (e.g., tornado, grand vista), and cognitive elicitors (e.g., grand theory).
Text change:
Line 27-32
Awe is an important research topic in positive psychology, which refers to an emotion when individuals encounter vast and powerful stimuli that are beyond their own understanding, and they should accommodate to the current context [2]. There are many kinds of elicitors that induce awe, such as social elicitors (e.g., powerful leader, religion), physical elicitors (e.g., tornado, grand vista), and cognitive elicitors (e.g., grand theory) [2].
Line 111-116
Although previous studies have explored the relationship among dispositional awe, connectedness, and prosocial behavior, No study has examined the serial mediation effect of connectedness and empathy. The exploration of this question could deepen our understanding the relationship between awe and prosocial tendencies. Therefore, the purpose of this study is to investigate the mechanisms by which dispositional awe predicts prosocial tendencies from the perspective of connectedness and empathy.
- The chapter “1. Introduction” is well written and gives the state of art to the research topic. References to previous literature are clear. Anyhow, I can recommend some recents articles and books about empathy and connecteddness.
Response:
Thanks for this recommendation. We have read the literature (Raatikainen et al., 2020) and added it in the manuscript at the lines of 262-263, 441-442. We are so sorry that the other two articles were not found. Therefore, we did not refer.
Text change:
Line 266-267
Previous studies showed that connectedness could promote one’s prosociality in many forms [26, 39, 40, 48-49].
- Materials and methods – I am not expert of evaluating this type of methodology and its presentation.
Response:
Thank you for your comments. In the present study, we used mature questionnaires, and described each measurement instrument according to the requirements of psychology. We also have revised this section according to the journal request and style.
Text change:
Line 125, 142, 149, 157, 171, 178
2.1 Study Sample and Data Collection
2.2 Materials
2.2.1 Dispositional awe
2.2.2 Prosocial tendencies
2.2.3 Connectedness
2.2.4 Empathy
2.2.5 Small self
- In the discussion the sentence: “Consistent with our findings, a recent study found that people with greater dispositional awe-proneness were more willingto self-sacrifice for their group.” This makes me again wonder, in what kind of society this is a desired pattern of behaviour. In army? Coud this refer to suicide bombe?
Response:
Thank you very much for this comment. In this article, the indexes of self-sacrifice for the group included monetary distribution, Trolley Dilemma and willingness to self-sacrifice for the religious organizations. The measure of willingness to self-sacrifice did include the probability of harming the out-group member. Therefore, to avoid misdirection, we have deleted this reference.
- In the paragraph starting from the line 234 you tell of positive impact of awe. But in my opinion, from line 242 you jump to impact that is not desired. Maladaptive narcissism is not hoped part of personality. This jump from good to bad should be somehow separated in the text.
Response:
Thank you very much to pointing this out. On the one hand, we think the impact of awe is positive in general. We want to express that awe could negatively predict maladaptive narcissism. Maybe the inaccurate expression caused your misunderstanding. We have corrected the inaccurate statement in the manuscript. On the other hand,we have corrected the statement that narcissism is one type of personalities.
Text change:
Line 269-272
Another study also has shown that connectedness could interpret the negative predictive effect of awe on maladaptive narcissism, a tendency of grandiose fantasies and the need for admiration which may reduce concern for others’ welfare [50].
- Conclusion could be a bit long and expalain again why this research was made and how.
Response:
Thank you very much for pointing this out. According to your suggestion, we have modified the conclusion.
Text change:
Line 310-319
Previous studies have examined the mechanism of awe on prosocial behavior from the perspective of small self, but have not found consistent results. Based on Feelings as information theory and Broaden-and-build theory of positive emotions, this work examined the mediation role of connectedness and empathy in awe on prosocial behavior. It is found that awe positively predicts the sense of connectedness with society and the world, which in turn links with empathic concern for other people and prosocial tendencies. These results exist independently even after controlling for the effect of small self. The present investigation reveals the impact of awe on prosocial tendencies and the mechanisms, which extends our understanding of the mechanisms underlying the relationship between awe and prosociality.
Possible reading
Raatikainen, K.J.; Juhola, K; Huhmarniemi, M. & Peña-Lagos, H. (2020) “Face the cow”: reconnecting to nature and increasing capacities for pro-environmental agency, Ecosystems and People, 16:1, 273-289, DOI: 10.1080/26395916.2020.1817151
Kravtsov, T.; Huhmarniemi, M. & Kugapi, O. (2022). Enhancing empathy in creative tourism. In M. Sarantou & S. Miettinen (Eds.), Empathy and business transformations (pp. 31–43 ). Routledge.
- Sarantou & S. Miettinen (Eds.) (2022). Empathy and business transformations https://doi.org/10.4324/9781003227557

Reviewer 3 Report
The main objective of this paper is to study the relationships between awe and prosocial behavior through serial mediation effects of the sense of connectedness and empathy after controlling for the effect of small self. Specifically, they propose that awe will influence the sense of connectedness, which in turn will impact on empathy, thereby enhancing prosocial tendencies. The results support that awe positively predict prosocial tendencies, and this result was partially mediated by connectedness and empathy, but not by the sense of small self.
I would like to highlight three of the strengths of this work. The first is the relevance and timeliness of the topic addressed. Identifying factors related to prosocial tendencies places this research at the core of many social problems of concern in today's societies and in the development of friendlier societies.
The perspective the paper adopts, therefore, has potential theoretical and applied implications.
Second, inquiring into the mechanisms underlying these effects and relationships has important theoretical and applied implications.
Second, the authors' arguments and predictions are solidly supported by evidence and work that justifies their relevance and pertinence, and the methodological treatment and analyses undertaken are appropriate to the stated objectives.
Finally, I would also like to highlight the style in which the text has been written. It is simple, clear and direct and maintains an organization that makes it easy to read and understand.
On the other hand, there are three issues that I would like to suggest to the authors.
1. It would be desirable to include a graphic of the proposed media model(optional)
2. It would be interesting to compare the strength of the indirect effects. Given that your main hypothesis is the mediational serial effect (awe - connectedness - empathy - prosocial tendencies) check whether these effects are superior to the indirect effects of the two mediating variables separately.
3. On the cognitive and affective mechanisms of empathy (optional)
The authors introduce several theories and hypotheses that could explain the effects of awe on prosocial behavior and the proposed mediational effects. Numerous research studies have shown that the cognitive and affective scales of the Interpersonal Reactivity Index (IRI) are differently related to prosocial behavior and the desire to help. It would be interesting to explore this perspective in the study of the mediational role of empathy, exploring whether mediational effects persist and the sense in which these two dimensions are related to prosocial behavior. If the authors do not consider it appropriate to include it in the analyses, it would be interesting to address it in the discussion.
4. Cite the reference from which the measurement instrument is drawn (line 131) Prosocial Tendency Measure (PTM)
Author Response
- It would be desirable to include a graphic of the proposed media model(optional)
Response:
Thank you very much for pointing this out. According to your suggestion, we have added a graphic of the proposed media model in the manuscript.
Text change:
Line 121-123
The proposed media model was shown in Figure 1.
Figure 1 The proposed media model
- It would be interesting to compare the strength of the indirect effects. Given that your main hypothesis is the mediational serial effect (awe - connectedness - empathy - prosocial tendencies) check whether these effects are superior to the indirect effects of the two mediating variables separately.
Response:
Thank you very much for this suggestion. According to this suggestion, we have analyzed data and compare the strength of the indirect effects. We found that that empathy was the strongest mediator between awe and each type of prosocial tendencies, and the serial mediation effect was the smallest. We have add this result in the manuscript.
Text change:
Line 232-239
Using a pairwise comparison of the strength of the three mediation effects, we found that the mediation effect of empathy was stronger than the serial mediation effect between awe and each type of prosocial tendencies, ps< .001, and the mediation effect of connectedness was stronger than the serial mediation effect between awe and prosocial tendencies under the contexts of altruism, anonymity, and emergency, ps< .018. In addition, the effect of empathy was stronger than that of connectedness under the public contexts p = .001. other pairwise comparisons were insignificant, p > .058.
- On the cognitive and affective mechanisms of empathy (optional)
The authors introduce several theories and hypotheses that could explain the effects of awe on prosocial behavior and the proposed mediational effects. Numerous research studies have shown that the cognitive and affective scales of the Interpersonal Reactivity Index (IRI) are differently related to prosocial behavior and the desire to help. It would be interesting to explore this perspective in the study of the mediational role of empathy, exploring whether mediational effects persist and the sense in which these two dimensions are related to prosocial behavior. If the authors do not consider it appropriate to include it in the analyses, it would be interesting to address it in the discussion.
Response:
Thank you very much for this enlightening advice. Since the purpose of the present study was to examine the mediation effect of general empathy between awe and prosocial tendencies, rather than the mediation effects of cognitive and affective empathy, we did not include it in the analyses. This suggestion is enlightening, so we discussed it in the section of limitations.
Text change:
Line 304-308
In addition, empathy is a multi-dimensional concept, including emotional empathy and cognitive empathy [54]. Previous studies have found that different components of empathy have different effects on prosocial behavior [55]. Therefore, whether cognitive and emotional empathy play different roles in the relationship between awe and prosocial behavior also needs further investigation.
References
- 55. Zaki, J. and K.N. Ochsner The neuroscience of empathy: progress, pitfalls and promise. Nature neuroscience, 2012. 15, 675-680 DOI: 10.1038/nn.3085.
- 56. Edele, A., I. Dziobek, and M. Keller, Explaining altruistic sharing in the dictator game: The role of affective empathy, cognitive empathy, and justice sensitivity.Learning and Individual Differences, 2013. 24, 96-102.
- Cite the reference from which the measurement instrument is drawn (line 131) Prosocial Tendency Measure (PTM)
Response:
Thanks very much for your careful examination. We have added this literature in the manuscript.
Text change:
Line 150
Prosocial Tendency Measure (PTM) was used to assess prosocial tendencies [42].
Line 430-431
- Carlo, G. and B.A. Randall, The Development of a Measure of Prosocial Behaviors for Late Adolescents.Journal of Youth and Adolescence, 2002. 31(1): p. 31-44.

Round 2
Reviewer 2 Report
Thank you for your detail work with editing the manuscript.